# ResBinNet: Residual Binary Neural Network

## Abstract

Recent efforts on training light-weight binary neural networks offer promising execution/memory efficiency. This paper introduces ResBinNet, which is a composition of two interlinked methodologies aiming to address the slow convergence speed and limited accuracy of binary convolutional neural networks. The first method, called residual binarization, learns a multi-level binary representation for the features within a certain neural network layer. The second method, called temperature adjustment, gradually binarizes the weights of a particular layer. The two methods jointly learn a set of soft-binarized parameters that improve the convergence rate and accuracy of binary neural networks. We corroborate the applicability and scalability of ResBinNet by implementing a prototype hardware accelerator. The accelerator is reconfigurable in terms of the numerical precision of the binarized features, offering a trade-off between runtime and inference accuracy.

## 1 Introduction

Convolutional Neural Networks (CNNs) have shown promising inference accuracy for learning applications in various domains. These models are generally over-parameterized to facilitate the convergence during the training phase (Hinton et al. (2012); Denil et al. (2013)). A line of optimization methodologies such as tensor decomposition (Kim et al. (2015); Zhang et al. (2015)), parameter quantization (Hubara et al. (2016); Han et al. (2015)), sparse convolutions (Liu et al. (2015); Wen et al. (2016)), and binary networks (Courbariaux et al. (2016); Rastegari et al. (2016)) have been proposed to reduce the complexity of neural networks for efficient execution. Among these works, binary neural networks result in two particular benefits: (i) They reduce the memory footprint by a factor of 32 compared to the full-precision model; this is specifically important since memory access plays an essential role in the execution of CNNs on resource-constrained devices. (ii) Binary networks replace the costly multiplications with simple XNOR operations (Rastegari et al. (2016); Umuroglu et al. (2017)), reducing the execution time and energy consumption significantly.

Considering the prior art, there exist two major challenges associated with binary neural networks. First, the convergence rate of the existing solutions for training binary CNNs is considerably slower than their full-precision counterparts. Second, in order to achieve comparable classification accuracy, binarized neural networks often compensate for the numerical precision loss by employing high dimensional feature maps in a wide CNN topology, which in turn reduces the effective compression rate. As a result, full-precision networks often surpass binary networks in terms of convergence rate and final achievable accuracy.

In this paper, we propose ResBinNet, a novel solution for increasing the convergence rate and the final accuracy of binary networks. The global flow of ResBinNet is depicted in Figure 1. The first phase, which we call *Soft Binarization*, includes two methodologies that we propose to address the aforementioned challenges for training binary CNNs. First, we introduce a *Residual Binarization* scheme which allows the number of possible values for activation units to be reconfigurable at runtime. To this purpose, we learn a multi-level residual representation for the features within the CNN to adaptively increase the numerical precision of the activation units. Second, we introduce a novel weight binarization approach, called *Tempreture Adjustment*, which aims to gradually enforce binarization constraints over the weight parameters throughout the training phase. The two interlinked methods significantly improve both the convergence rate and the final accuracy of ResBinNet compared to prior art. Once the soft training phase is finished, we convert the weights to actual binary values (0,1). Fine-tuning of the model is then performed in *Hard Binarization* phase using existing training algorithms (e.g. BinaryNets (Courbariaux et al. (2016))) in few epochs (e.g. one epoch).

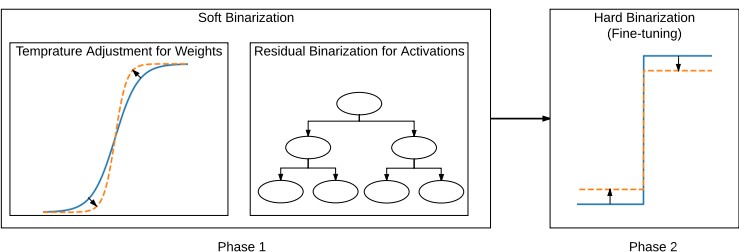

Figure 1: The global flow of ResBinNet binary training. Residual binarization learns a multi-level representation for feature maps. Temperature adjustment performs a change of variable over the trainable weights and gradually pushes them towards binary values during the training phase.

ResBinNet is designed to fulfill certain goals: (i) It should enable reconfigurability for binary neural networks; in other words, the number of residual binary representatives should be adjustable to offer a trade-off between inference accuracy and computation time. (ii) The multi-level binarized features should be compatible with the XNOR multiplication approach proposed in the existing literature. (iii) ResBinNet should speed up the convergence rate of binarized CNNs. (iv) Current hardware accelerators for binary CNNs should be able to benefit from ResBinNet with minimum modification in their design. In summary, the contributions of this paper are as follows:

- Proposing residual binarization, a methodology for learning multi-level residual representations for each feature map in binary CNNs.

- Introducing temperature adjustment as a practical approach for gradual (soft) binarization of CNN weights.

- Analyzing the trade-off between accuracy and execution time of ResBinNet on a real hardware design.

- Evaluating ResBinNet convergence rate and accuracy on three datasets: MNIST, SVHN, and CIFAR-10.

- Development of an open-source Application Program Interface (API) for ResBinNet[1].

The remainder of the paper is organized as follows: In Section 2, we describe the residual binarization method for binarizing activations. Section 3 explains the temperature adjustment technique for binarizing weights. In Section 4, we discuss how particular ResBinNet operations (e.g. multi-level XNOR-popcount) can be efficiently implemented on existing hardware accelerators. Experiments are discussed in Section 5. Finally, we discuss the related work and conclusion in Sections 6 and 7.

## 2 RESIDUAL BINARIZATION

A binarization scheme converts value $x$ to the binarized estimation $e_x$, which can take one of the possible values $\gamma$ or $-\gamma$. This representation allows us to represent $e_x$ with a single bit $b_x$. In particular, for a given layer within the CNN, we can store the single full-precision value of $\gamma$ as a representative for all features, and reduce the memory footprint by storing bits $b_x$ instead of $x$ for each feature. Assuming that both the weights and input features of a CNN layer are binarized, each dot product between a feature vector $\vec{x}$ and weight vector $\vec{w}$ can be efficiently computed using XNOR-popcount operations as previously suggested in (Courbariaux et al. (2016); Rastegari et al. (2016)). Let $\vec{x} = \gamma_x \, \vec{s}_x$ and $\vec{w} = \gamma_w \, \vec{s}_w$ where $\{\gamma_x, \gamma_w\}$ are scalar values and $\{\vec{s}_x, \vec{s}_w\}$ are the corresponding sign vectors. The binary representations of $\{\vec{x}, \vec{y}\}$, which we denote by $\{\vec{b}_x, \vec{b}_w\}$, are simply computed by encoding the sign vectors $\vec{s}$ to binary vectors. The dot product between $\vec{x}$ and $\vec{w}$ can be computed as:

$$dot(\vec{w}, \vec{x}) \; = \gamma_x \gamma_w \, dot(\vec{s}_x, \vec{s}_w) = \gamma_x \gamma_w \, xnorpopcount(\vec{b_x}, \vec{b_w}),$$

where $xnorpopcount(. \, , .)$ returns the number of set bits in the element-wise XNOR of the input binary vectors.

---

[1]link omitted due to the anonymous review process

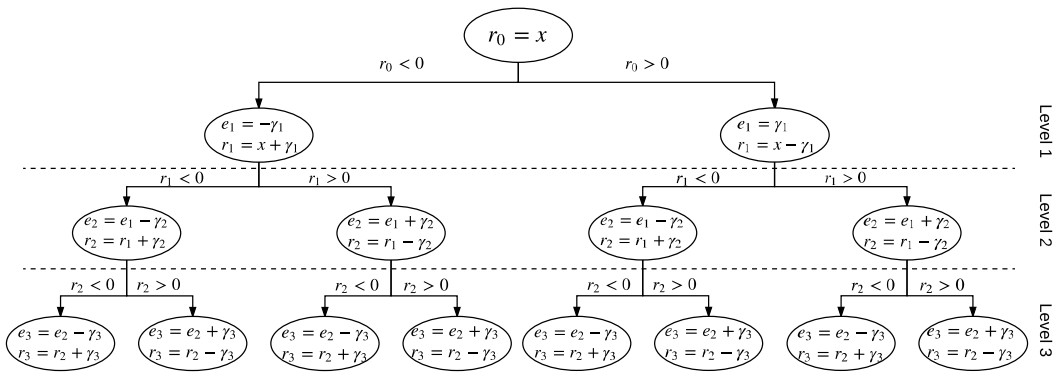

Figure 2: Schematic flow for computing 3 levels of residual binary estimates $e$. As we go deeper in levels, the estimation becomes more accurate.

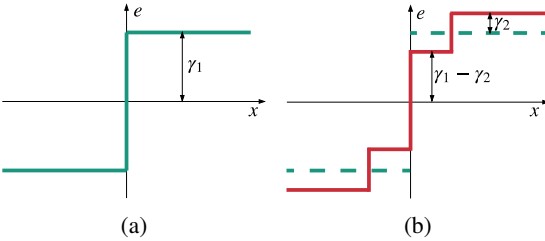

Figure 3: Illustration of binarized activation function. (a) Conventional 1-level binarization. (b) Residual binarization with two levels.

**Multi-level Residual Binarization**: Imposing binary constraints on weights and activations of a neural network inherently limits the model's ability to provide the inference accuracy that a floating-point counterpart can achieve. To address this issue, we propose a multi-level binarization scheme where the residual errors are sequentially binarized to increase the numerical precision of the estimation. Figure 2 presents the procedure to compute an estimate $e$ from input $x$. Each level of the graph (say the $i^{th}$ level) computes the corresponding estimate $e_i$ by taking the sign of its input (the residual error from the $(i-1)^{th}$ level), multiplying it by a parameter $\gamma_i$, and adding $\pm\gamma_i$ to the estimate of the previous level. In addition, it computes the residual error $r_i$ and feeds it to the input of the next level. The estimates of deeper levels are therefore more accurate representations for the input $x$. Note that the estimate $e_i$ in level $i$ can be represented using a stream of $i$ bits corresponding to the signs of inputs to the first $i$ levels.

**Residual Binary Activation Function**: Similar to previous works which use the *Sign* function as the activation function, in this paper we use the residual binarization to account for the activation function. The difference between our approach and the single-bit approach is shown in Figure 3. Each level has a separate full-precision representative $\gamma_i$, which should be learned in the training phase. In this setting, the gradients in the backward propagation are computed the same way as in the conventional single-bit binarization (Courbariaux et al. (2016); Rastegari et al. (2016)), regardless of the number of residual levels. In the forward propagation, however, the computed results of our approach provide a more accurate approximation. For instance, if we employ 2 levels of residual binarization, the activation function can take $2^2 = 4$ different values. In general, the total number of possible values for the activation functions for an $l$-level residual binarization scheme is $2^l$.

**Multi-level XNOR-popcount**: In ResBinNet, the dot product of an $l$-level residual-binarized feature vector $\vec{e}$ and a vector of binary weights $\vec{w}$ can be rendered using $l$ subsequent XNOR-popcount operations. Let $\vec{e} = \sum_{i=1}^{l} \gamma_{ei}\vec{s}_{ei}$ and $\vec{w} = \gamma_w\vec{s}_w$, where $\vec{s}_{ei}$ and $\vec{s}_w$ correspond to the sign of $i^{th}$ residual in $\vec{e}$ and sign of $\vec{w}$, respectively. The dot product between $\vec{e}$ and $\vec{w}$ is computed as:

$$dot(\vec{w}, \vec{e}) = dot(\sum_{i=1}^{l} \gamma_{ei}\vec{s}_{ei}, \gamma_w\vec{s}_w) = \sum_{i=1}^{l} \gamma_{ei}\gamma_w \, dot(\vec{s}_{ei}, \vec{s}_w) = \sum_{i=1}^{l} \gamma_{ei}\gamma_w \, xnorpopcount(\vec{b}_{ei}, \vec{b}_w),$$

where $\{\vec{b}_{ei}, \vec{b}_w\}$ are the binary representations corresponding to $\{\vec{s}_{ei}, \vec{s}_w\}$, respectively. Note that the subsequent XNOR-popcount operations can be performed sequentially, thus, the same memory used for operating on $\vec{b}_{ei}$ can be reused for operating on $\vec{b}_{ei+1}$. As a result, the actual memory footprint for a multi-level residual binarization is the same as that of a single-level binarization, provided that the bit streams are processed sequentially.

**Residual Encoding**: In order to convert matrix-vector multiplications into XNOR-popcount operations, we need to encode a feature $x$ into a stream of binary values $\{b_{ei}|i \in 1, 2, \dots, l\}$. The pseudo code for this operation, which we call *Residual Encoding*, is presented in Algorithm 1.

---

**Algorithm 1** $l$-level residual encoding algorithm

**inputs:** $\gamma_1, \gamma_2, ..., \gamma_l, x$
**outputs:** $b_{e1}, b_{e2}, ..., b_{el}$

1: $r \leftarrow x$
2: $e \leftarrow 0$
3: **for** $i = 1 \dots l$ **do**
4: $\quad b_{ei} \leftarrow Binarize(Sign(r))$
5: $\quad e \leftarrow e + Sign(r) \times \gamma_i$
6: $\quad r \leftarrow r - Sign(r) \times \gamma_i$
7: **end for**

---

## 3 TEMPERATURE ADJUSTMENT

Approximating the weights of a neural network with binary values often results in loss of accuracy in the pertinent model. In this section, we explain our methodology to minimize the approximation error during the training, such that the trained weights exhibit lower binarization errors. Let $W$ denote the parameter set within a certain layer of the neural network. Instead of directly using $W$ to compute the layer's output, we perform a change of variable $\theta = \gamma\, H(\alpha W)$ and compute the output using $\theta$. Here, $H(.)$ is a bounded, monotonically-increasing function such as the *Hyperbolic Tangent* function that is applied on $W$ element-wise. Parameter $\gamma$ is a trainable parameter that adjusts the maximum and minimum values that $\theta$ can take. Parameter $\alpha$, which we call the *Temperature* henceforth, controls the slope of function $H(.)$.

**Effect on Binarization**: Figure 4a and 4b illustrate the effect of parameters $\alpha$ and $\gamma$ on the nonlinear change of variable $\theta = \gamma\, Tanh(\alpha W)$. Note that $\theta$ acts as a semi-binarized parameter set in the soft training phase. As we increase the temperature parameter, $H(.)$ becomes closer to the binary *sign* function, meaning that the pertinent $\theta$ will exhibit less error when approximated with $\pm\gamma$. Note that $W$ and $\gamma$ are the trainable parameters in this setting. Parameter $\theta$ is used in the forward propagation phase of the soft training, while in the backward propagation step $W$ is updated.

**Effect on Training**: Let $g_\theta$ and $g_W$ be the gradients of the training loss function with respect to $\theta$ and $W$, respectively, then we have $g_W = g_\theta \times \frac{\partial\theta}{\partial W}$. In other words, the magnitude of the gradient that actually flows through $W$ is controlled by $\frac{\partial\theta}{\partial W}$. If $\theta$ is close to $\pm\gamma$, the gradient will be filtered out; otherwise, the gradients will flow through $W$.

**Effect of the Temperature on Gradients**: Figure 4c illustrates how the temperature parameter can affect the gradient filtering term $\frac{\partial\theta}{\partial W}$ during the training. As we increase the temperature, elements of $W$ that are closer to 0 receive amplified gradients, while the elements that are closer to the binary regime (i.e. $\theta \approx \pm\gamma$) encounter damped gradients. This means that increasing the temperature parameter $\alpha$ will push most of the weights to the binary regime with a bigger force; therefore, a neural network trained with high temperature values exhibits a smaller binarization error.

**Temperature Adjustment**: Setting a high temperature at the beginning of the training will eliminate most of the gradients, preventing the training loss from being optimized. To address this problem, we start the soft binarization phase with a low temperature (e.g. $\alpha = 1$) and slightly increase it at the end of each mini-batch. This approach gradually adapts the weights to binary values during the training. Figure 5 presents an example of the histogram of the semi-binarized weights $\theta$ in different training epochs. As can be seen, the distribution is gradually shifted towards binary values as the training proceeds. After soft binarization, the parameter set $\theta$ can be used as an initial point for

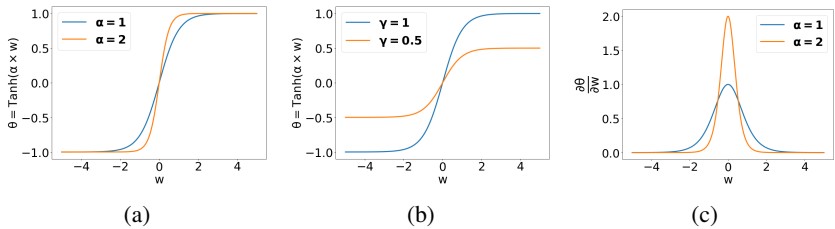

Figure 4: An example change of variable with *Tanh* nonlinearity. (a) The effect of the temperature parameter: higher $\alpha$ values provide better soft-binary estimations. (b) The effect of the bounding parameter: $\gamma$ is a trainable value for each weight matrix $W$. (c) The effect of the temperature parameter $\alpha$ on the gradient filtering term $\frac{\partial \theta}{\partial W}$.

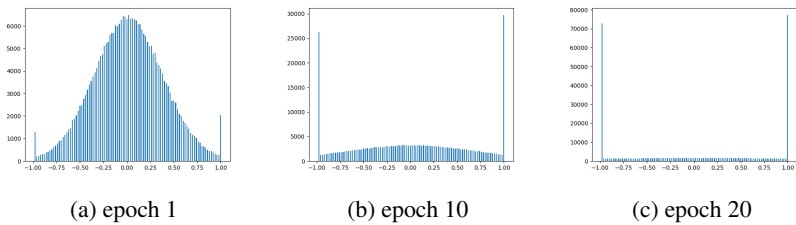

(a) epoch 1        (b) epoch 10        (c) epoch 20

Figure 5: Histogram of the elements of $\theta$ in a certain layer of the neural network during training.

existing hard binarization schemes such as the method proposed by (Courbariaux et al. (2016)). As We illustrate in Section 5, the soft binarization methodology significantly increases the convergence rate of binary CNNs.

## 4    HARDWARE ACCELERATOR MODULES

In this section, we show that the modifications required to incorporate residual binarization into existing hardware accelerators for binary CNNs are minimal. ResBinNet provides a trade-off between inference accuracy and the execution time, while keeping the implementation cost (e.g area cost) almost intact; as a result, ResBinNet readily offers users a decision on the latency of their learning application by compromising the inference accuracy.

As an example, we consider the FPGA accelerator for binary CNNs proposed by (Umuroglu et al. (2017)). We refer the reader to the mentioned paper for details about the original design. Here, we describe the modifications that we integrated into the specific components of their design to accommodate residual binarization. The modified accelerator will be publicly available on Github [2]. Figure 6 depicts a schematic view of the original hardware and our modified accelerator. Note that in the original implementation, each layer takes a single binary vector $\vec{b}_{in}$ and computes a single output vector $\vec{b}_{out}$ while the modified version processes $l$ streams of binary vectors where $l$ is the desired number of residual levels.

**Matrix-Vector Multiplication**: Both in the original and the modified accelerators, the matrix-vector multiplication unit is the most computationally intensive among all other operations. In the original design, this unit takes a binary vector $\vec{b}_{in}$ and outputs a full-precision vector $\vec{y}$. To accommodate residual binarization, we modify this module as follows: the XNOR-popcount operation is sequentially performed on the stream of binary vectors $\vec{b}_{in,i}$. Each XNOR-popcount results in a different vector $\vec{y}_i$. Then, the output is computed as $\vec{y} = \sum_i \gamma_i \vec{y}_i$. Note that the computation overhead of the summation is negligible compared to the XNOR-popcount operation, thus, the runtime of multi-level XNOR-popcount with $l$-level residual representations is approximately $l$ times the runtime of the conventional XNOR-popcount in the original design.

---

[2]link omitted due to the anonymous review process

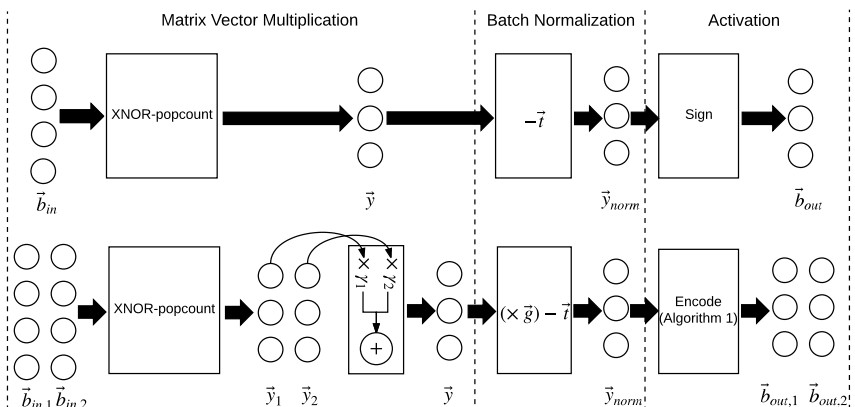

Figure 6: Hardware architecture of the baseline (top) and our modified (bottom) binary CNN layer.

**Batch-Normalization and Activation Function**: Batch-Normalization in the inference phase can be viewed as multiplying a vector $\vec{y}$ by constant vector $\vec{g}$ and subtracting vector $\vec{t}$ to obtain the normalized vector $\vec{y}_{norm}$. The original design in (Umuroglu et al. (2017)) does not require the multiplication step since only the sign of $\vec{y}_{norm}$ matters to compute the output of the activation function (the Sign function). In our design, the multiplication step is necessary since the value of $\vec{y}_{norm}$ affects the output of our activation function, which is encoded using Algorithm 1 and sent to the next layer to be used as an input.

**Max-Pooling**: The original implementation simply computes the Boolean OR of the binary values to perform max pooling over the features within a window. In ResBinNet, however, features are represented with $l$-bit residual representations. As a result, performing Boolean OR over the binary encodings is no longer equivalent to performing max-pooling over the features. Nevertheless, the pooling operation can be performed over the encoded values directly. Assume full-precision values $e_x$ and $e_y$, with $l$-level binary encodings $b_x$ and $b_y$, respectively. Considering ordered positive $\gamma_i$ values (i.e. $\gamma_1 > \gamma_2 > \ldots > \gamma_l > 0$), we can easily conclude that if $e_x < e_y$ then $b_x < b_y$.

## 5 Experiments

We implement our API using Keras (Chollet et al. (2015)) library with a Tensorflow backend. The synthesis reports (resource utilization and latency) for the FPGA accelerator are gathered using Vivado Design Suite (Xilinx (2017)). For temperature adjustment (Section 3), we use a "hard tanh" nonlinearity and gradually increase the temperature by incrementing $\alpha$ at the end of each epoch.

We evaluate ResBinNet by comparing the accuracy, number of training epochs, size of the network, and execution time. Proof-of-concept evaluations are performed for three datasets: MNIST, CIFAR-10, and SVHN. Table 1 presents the architecture of the trained neural networks. The architectures are picked from (Umuroglu et al. (2017)).

Table 1: Network architectures for evaluation benchmarks. $C64$ denotes a $3 \times 3$ convolution with 64 output channels, $MP$ stands for $2 \times 2$ max pooling, $BN$ represents batch normalization, and $D512$ means a dense layer with 512 outputs. The residual binarizations are shown using $RB$.

| Benchmark | CNN Architecture |
|---|---|
| MNIST | $784\,(input) - D256 - BN - RB - D256 - BN-$ $RB - D256 - BN - RB - D10 - BN - Softmax$ |
| CIFAR10 & SVHN | $3 \times 32 \times 32\,(input) - C64 - BN - RB - C64-$ $BN - RB - MP - C128 - BN - RB - C128-$ $BN - RB - MP - C256 - BN - RB - C256-$ $BN - RB - D512 - BN - RB - D512 - BN-$ $RB - D10 - BN - Softmax$ |

**Effect of the model size on accuracy**: As discussed in (Umuroglu et al. (2017)), the final accuracy of the binary CNN for a particular application is correlated with the shape of the network. For

Table 2: Comparison of model size, number of training epochs, and accuracy.

| Benchmark | Binarynet (Courbariaux et al. (2016)) | | | FINN (Umuroglu et al. (2017)) | | | ResBinNet | | | | |
|---|---|---|---|---|---|---|---|---|---|---|---|
| | # Epochs | Accuracy | Size (Mbits) | # Epochs | Accuracy | Size (Mbits) | # Epochs | Size (Mbits) | Accuracy (1-level) | Accuracy (2-level) | Accuracy (3-level) |
| CIFAR-10 | 500 | 89.85% | 5.24 | NA | 80.1% | 1.5 | 50+1 | 1.5 | 76% | 83.5% | 84.6% |
| SVHN | 200 | 97.47% | 5.24 | NA | 94.9% | 1.5 | 10+1 | 1.5 | 95.2% | 96.9% | 97.1% |
| MNIST | 1000 | 99.04% | 52.7 | NA | 95.83% | 0.3 | 30+1 | 0.3 | 97.3% | 97.9% | 98.1% |

instance, authors of the paper report that the accuracy of MNIST for the architecture in Table 2 varies in the range (95.83%-98.4%) when the number of neurons in hidden layers is varied from 256 to 1024. Similarly, the architecture in (Umuroglu et al. (2017)) for CIFAR-10 is a smaller version of the architecture originally trained by (Courbariaux et al. (2016)). Using this smaller architecture drops the accuracy from 88.6% to 80.1%. In our evaluations, we show that ResBinNet can reduce the accuracy drop using more residual binary levels for the *activations* of the smaller model.

**Effect of the number of epochs on accuracy**: Compared to full-precision neural networks, binarized CNNs usually need more training epochs to achieve the same accuracy. For example, the CIFAR-10 architecture in (Courbariaux et al. (2016)) is trained for 500 epochs, while the full-precision version of the same network can achieve comparable accuracy in roughly 50 iterations [3]. Here, we argue that soft binarization in ResBinNet can significantly reduce the number of training epochs for binary CNNs.

Table 2 compares ResBinNet with two prior arts, namely Binarynet and FINN. Both baselines use the same training methodology, but the network architectures in FINN are considerably smaller, which leads to lower accuracy rates for FINN. We evaluate ResBinNet using the small architectures of FINN. The training of ResBinNet consists of a soft binarization phase and a single fine-tuning epoch. Note that the fine-tuning phase uses the same algorithm as the two baselines. The higher accuracy of Binarynet compared to our approach is a direct result of employing a large architecture and training for many epochs. For each benchmark, the comparison between our approach and the same network architecture of FINN is followed:

- **CIFAR-10**: Compared to FINN, ResBinNet achieves higher accuracy for more than 1 level of residual binarization. We argue that, even for 1-level binarization, the same accuracy is viable if we fine-tune the soft-binarized model (after 50 epochs) for more than 1 epochs (FINN and ResBinNet use the same algorithms in this phase). In addition, the convergence rate of ResBinNet is improved as the number of residual levels is increased.

- **SVHN and MNIST**: For these datasets, ResBinNet achieves a higher accuracy with even fewer epochs compared to CIFAR-10. The final accuracy and the convergence speed also exhibit improvement as the number of residual levels is increased from 1 to 3.

We now evaluate the area overhead and execution time of ResBinNet for the modified hardware architecture, which we discussed previously in Section 4. We compare the implementation of the CNN architecture used for the CIFAR-10 and SVHN tasks (See Table 1). Figure 7 compares the hardware resource utilization, and execution time per input.

The resource utilization of ResBinNet is evaluated in Figure 7a, which compares the utilization (in %) for different resources of the FPGA (i.e. BRAM, DSP, LUT, and Registers). For each resource, we compare the baseline with different number of residual binarization levels in ResBinNet. Asides from the DSP utilization, which is required for full-precision multiplications in batch normalization, the other three resources show a modest increase in utilization, meaning that the residual binarization method offers a scalable design for real-world systems.

Figure 7b compares the latency (runtime) of ResBinNet with the baseline accelerator. In particular, we consider multi-level residual binarization with 1, 2, and 3 residual levels which are denoted by RBN1, RBN2, and RBN3, respectively. The numbers on top of the bars show the accuracy of the corresponding binarized CNN for CIFAR-10 task. As can be seen, ResBinNet enables users to achieve higher accuracy by tolerating a higher latency, which is almost linear with respect to the number of residual levels.

---

[3]We trained the full precision network and validated this fact

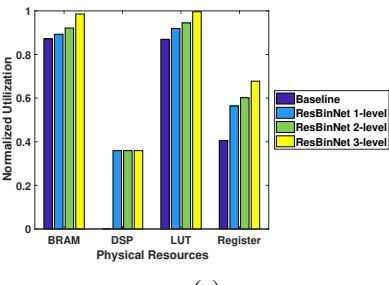 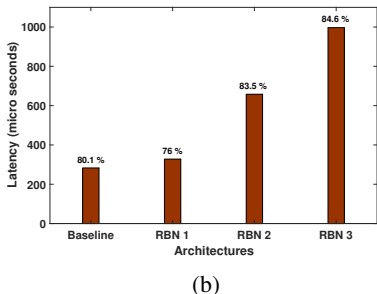

(a)                                      (b)

Figure 7: (a) Resource utilization overhead of ResBinNet with different residual levels versus baseline design( Umuroglu et al. (2017)) implemented on Xilinx ZC706 Evaluation Kit. (b) Latency-accuracy trade-off offered by ResBinNet with different residual levels.

## 6 RELATED WORK

Training CNNs with binary weights and/or activations has been the subject of very recent works (Courbariaux et al. (2015); Rastegari et al. (2016); Courbariaux et al. (2016); Umuroglu et al. (2017)). The authors of Binaryconnect (Courbariaux et al. (2015)) suggest a probabilistic methodology that leverages the full-precision weights to generate binary representatives during forward pass while in the back-propagation the full-precision weights are updated. (Courbariaux et al. (2016)) is the first work attempting to binarize both weight and activations of CNN. In this work, authors also suggest replacing the costly dot products by XNOR-popcount operations. XNOR-net (Rastegari et al. (2016)) proposes to use scale factors during training, which results in an improved accuracy. The aforementioned works propose optimization solutions that enable the use of binarized values in CNNs which, in turn, enable the design of simple and efficient hardware accelerators. The downside of these works is that, aside from changing the architecture of the CNN (Umuroglu et al. (2017)), they do not offer any other reconfigurability in their designs.

On another track of research, the reconfigurability of CNN accelerators has been investigated. This line of research focuses on using adaptive low bit-width representations for compressing the parameters and/or simplifying the pertinent arithmetic operations (Zhou et al. (2016); Han et al. (2016); Wu et al. (2016); Cai et al. (2017)). The proposed solutions, however, do not enjoy the same simplified XNOR-popcount operations as in binarized CNNs.

Among the aforementioned works, a unified solution which is both reconfigurable and binarized is missing. To the best of our knowledge, ResBinNet is the first to offer a solution which is reconfigurable and, at the same time, enjoys the benefits of binarized CNNs. Our goal in the design of ResBinNet was to remain consistent with the existing CNN optimization solutions. As shown in the paper, ResBinNet is compatible with the accelerators designed for binarized CNNs.

## 7 CONCLUSION

This paper introduces ResBinNet, a novel reconfigurable binarization scheme which aims to improve the convergence rate and the final accuracy of binary CNNs. The proposed training is twofold: (i) In the first phase, called soft binarization, we introduce two distinct methodologies designed for binarizing weights and feature within CNNs, namely residual binarization, and temperature adjustment. Residual binarization learns a multi-level representation for features of CNN to provide an arbitrary numerical precision during inference. Temperature adjustment gradually imposes binarization constraints on the weights. (ii) In the second phase, which we call hard binarization, the model is fine-tuned in few training epochs. Our experiments demonstrate that the joint use of residual binarization and temperature adjustment improves the convergence rate and the accuracy of the binarized CNN. We argue that ResBinNet methodology can be adopted by current CNN hardware accelerators as it requires minimal modification to existing binarized CNN solutions. Developers can integrate the approaches proposed in this paper into their deep learning systems to provide users with a trade-off between application latency and inference accuracy.

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
