# OpenReview forum: "ResBinNet: Residual Binary Neural Network"
_ICLR.cc/2018/Conference — Reject_

### Official Review · AnonReviewer1 · 2017-11-24
**Experimental results seem weak**

**Rating:** 4
**Confidence:** 4

**Review:**

This paper proposes a method to quantize weights and activations in neural network during propagations.

The residual binarization idea is interesting. However, the experimental results are not sufficiently convincing that this method is meaningfully improving over previous methods. Specifically:

1) In table 2, the 1-st level method is not performing better the FINN, while at the higher levels we pay with a much higher latency (about x2-x3 in figure 7) to get slightly better accuracy.

2) Even at the highest level, the proposed method is not performing better than BinaryNet in terms of accuracy. The only gain in this comparison is the number of epochs needed for training. However, this is might be due to the size difference between the models, and not due to the proposed method.

3) In a comment during the review period, the authors mention that "For Imagenet, we can obtain a top-1 accuracy of 28.4%, 32.6%, and 33.6% for an Alexnet architecture with  1-3 levels of residual binarizations, while the Binarynet baseline achieves a top-1 accuracy of 27.9% with the same architecture." However, this is not accurate, BinaryNet actually achieves 41.8% top-1 accuracy for Imagenet with Alexnet (e.g., see BNN on table 2 in Hubara et al.).

Minor comment regarding novelty:
The temperature adjustment method sounds somewhat similar to previous method of increasing the slope described "Adjustable Bounded Rectifiers: Towards Deep Binary Representations"

---

### Official Review · AnonReviewer3 · 2017-11-27
**The experimental comparison are not enough**

**Rating:** 4
**Confidence:** 4

**Review:**

1. The idea of multi-level binarization is not new. The author may have a check at  Section "Multiple binarizations" in [a] and Section 3.1 in [b]. The author should also have a discussion on these works.

2. For the second contribution, the authors claim "Temperature Adjustment" significantly improves the convergence speed. This argument is not well supported by the experiments.

I prefer to see two plots: one for Binarynet and one for the proposed method. In these plot, testing accuracy v.s. the number of epoch (or time) should be shown. The total number of epochs in Table 2 does not tell anything.

3. Confusing in Table 2. In ResBinNet, why 1-, 2- and 3- level have the same size? Should more bits required by using higher level?

4. While the performance of the 1-bit system is not good, we can get very good results with 2 bits [a, c]. So, please also include [c] in the experimental comparison.

5. The proposed method can be trained end-to-end. However, a comparison with [b], which is a post-processing method, is still needed (see Question 1).

6. Could the authors also validate their proposed method on ImageNet? It is better to include GoogleNet and ResNet as well.

7. Could the authors make tables and figures in the experiment section large? It is hard to read in current size.

Reference
[a] How to Train a Compact Binary Neural Network with High Accuracy. AAAI 2017
[b] Network Sketching: Exploiting Binary Structure in Deep CNNs. CVPR 2017
[c] Trained Ternary Quantization. ICLR 2017

---

### Official Review · AnonReviewer2 · 2017-11-27
**Interesting work**

**Rating:** 4
**Confidence:** 4

**Review:**

This paper proposes ResBinNet, with residual binarization, and temperature adjustment. It is a reconfigurable binarization method for neural networks. It improves the convergence rate during training.

I appreciate a lot that the authors were able to validate their idea by building a prototype of an actual hardware accelerator.

I am wondering what are the values of \gamma’s in the residual binarization after learning? What is its advantage over having only one \gamma, and then the rest are just 1/2*\gamma, 1/4* \gamma, … , etc.? The latter is an important baseline for residual binarization because that corresponds to the widely used fixed point format for real numbers. If you can show some results that residual encoding is better than having {\gamma, 1/2*\gamma, 1/4* \gamma, …, } (which contains only one \gamma), it would validate the need of using this relatively complex binarization scheme. Otherwise, we can just use the l-bit fixed point multiplications, which is off-the-shelf and already highly optimized in many hardwares.

For the temperature adjustment, modifying the tanh() scale has already had a long history, for example, http://yann.lecun.com/exdb/publis/pdf/lecun-89.pdf page 7, which is exactly the same form as in this paper. Adjusting the slope during training has also been explored in some straight-through estimator approaches, such as https://arxiv.org/pdf/1609.01704.pdf. In addition, having this residual binarization and adjustable tanh(), is already adding extra computations for training. Could you provide some data for comparing the computations before and after adding residual binarization and temperature adjustment?

The authors claimed that ResBinNet converges faster during training, and in table 2 it shows that ResBinNet just needs 1/10 of the training epochs needed by BinaryNet. However, I don’t find it very fair. Given that the accuracy RBN gets is much lower than Binary Net, the readers might suspect that maybe the other two models already reach ResBinNet’s accuracy at an earlier training epochs (like epoch 50), and just take all the remaining epochs to reach a higher accuracy. On the other hand, this comparison is not fair for ResBinNet as well. The model size was much larger in BinaryNet than in ResBinNet. So it makes sense to train a BinaryNet or FINN, in the same size, and then compare the training curves. Lastly, in CIFAR-10 1-level case, it didn’t outperform FINN, which has the same size. Given these experiments, I can’t draw any convincing conclusion.

Apart from that, There is an error in Figure 7 (b), where the baseline has an accuracy of 80.1% but its corresponding bar is lower than RBN1, which has an accuracy of 76%.

---

### Public Comment · (anonymous) · 2017-11-08
**Difference with XNOR-net**

Considering the approach of XNOR-net (Rastegari et al. (2016)), what are the differences between your binarization and theirs? In particular, how are the Gamma values discussed in your paper different from the scaling factors used in XNOR-net?

In addition, I am curious how ResBinNet would compare to other works on large-scale tasks such as
Imagenet classification. Do you have any results in that direction?

Unless I am mistaken, the output of your “soft-binarization” method is a full-precision network. I am
wondering if the final “hard-binarization” can recover the accuracy of the full-precision model after re-
training. It would be better if you also reported the accuracy of the full-precision model (the soft-
binarized model) in Table 2.

---

> ### Author Response · Authors · 2017-11-09
> **Re: Difference with XNOR-net**
>
> Thank you very much for your comments. Here are the responses to your questions:
>
> Question1: We will emphasize the differences in the updated paper. In summary, the differences between our approach and XNOR-net are the following:
>
> - In XNOR-net, each layer utilizes multiple scaling factors for the weights. For example, it uses a separate scaling factor for each column of the weight matrix in a fully connected layer. In our approach,  the whole parameter set in one layer has a single Gamma value. This is particularly important to devise efficient hardware accelerators for the corresponding binary CNN.
>
> - Regarding the scaling factors for the activations, XNOR-net again uses multiple values for a certain layer. In our approach, the number of Gamma values for each layer is limited to the number of residual levels which is less than 4 in our experiments.
>
> - In XNOR-net, the scaling factors for the activations are dynamically computed during the execution by taking the average of feature maps. Computing these scaling factors involves a lot of full-precision operations which is in contrast with the whole rational of network binarization. In our approach, the Gamma values are learned in the training phase and they are fixed during the inference.
>
> All in all, the previous properties of XNOR-net help their design to achieve a higher accuracy, but prevents an efficient implementation of their binary network. To the best of our knowledge, no hardware accelerator has been proposed for XNOR-net.
>
> Question 2: The experiments in the paper aim to demonstrate the effectiveness of the approach. We have evaluated the method on Imagenet and will include the results in the revised version. For Imagenet, we can obtain a top-1 accuracy of 28.4%, 32.6%, and 33.6% for an Alexnet architecture with  1-3 levels of residual binarizations, while the Binarynet baseline achieves a top-1 accuracy of 27.9% with the same architecture.
>
> Question 3: The output of soft-binarization is actually full-precision; the point is that these full-precision values are so close to the binary values that the accuracy does not degrade significantly after hard-binarization. Note that we retrain the hard-binarized model for only 1 epoch. We will add the soft-binarized network accuracies in Table 2 as suggested.
>
> Hope the responses above clarified your questions.

---

### Decision · Program_Chairs · 2018-01-29
**ICLR 2018 Conference Acceptance Decision**

**Decision:**

Reject

**Comment:**

R1 and R3’s  main concern was that the work was not actually outperforming existing work and therefore its advantages were unclear. R2 brought up several questions on the experiments and asked for clarification with respect to previous work. R3 had several other detailed questions for the authors. The authors did not provide a response.